# Logistics Service Selection Strategy of Green Manufacturers in Green Low-Carbon Supply Chain

**DOI:** 10.3390/ijerph20043356

**Published:** 2023-02-14

**Authors:** Xigang Yuan, Wenwen Sheng, Min Wang, Dalin Zhang

**Affiliations:** 1Business School, Jiangsu Normal University, Xuzhou 221116, China; 2School of Business, Linyi University, Linyi 276000, China; 3Department of Computer Science, Aalborg University, 9220 Aalborgost, Denmark

**Keywords:** green low-carbon supply chain, e-commerce platform, logistics service mode, selling mode

## Abstract

In order to analyze the logistics service mode and sales mode selection strategy, a green low-carbon supply chain consisting of a single manufacturer and a single e-commerce platform is considered. First, in the green low-carbon supply chain which consists of a direct selling channel and reselling channel, the selection strategy of the manufacturer’s logistics service mode is analyzed. Second, in the green low-carbon supply chain which consists of a direct selling channel and an agency channel, the selection strategy of the manufacturer’s logistics service mode is analyzed. Last, the manufacturer’s sales mode is analyzed. We use the backward induction method to solve the theoretical model. This study contributes to the literature by considering the optimal decision of a green low-carbon supply chain. This study brings together the literature from streams on the selling channel selection strategy in green supply chains and the logistics service strategy in green supply chains. The impacts of the logistics service cost, the selling cost, and the green input cost coefficient on the optimal decision and the firms’ profit are discussed. The result shows that in the direct selling channel and reselling channel, when the basic market demand and the logistics service level of the third-party logistics service provider are low, manufacturers will choose the e-commerce platform logistics service; in the opposite case, manufacturers will choose the third-party logistics service. In the direct selling channel and agency channels, when the logistics service level of the third-party logistics service provider is greater than or equal to a certain critical value and less than or equal to the logistics service level of the e-commerce platform, manufacturers will choose the e-commerce platform logistics service; in the opposite case, manufacturers will choose the third-party logistics service. No matter whether the manufacturer chooses the logistics service provided by the third-party logistics service provider or the logistics service provided by the e-commerce platform, the manufacturer should choose the direct selling channel and the agency channel.

## 1. Introduction

According to “The Statistical Report on the Development of the Internet in China”, online selling in China reached CNY 11.76 trillion in 2020, an increase of 10.9% compared with 2019. Online retail selling of physical goods have reached CNY 9.76 trillion, accounting for 24.9% of the total retail selling of consumer goods. In December 2020, the number of Chinese online shopping users reached 782 million, representing an increase of 72.15 million compared with March 2020, accounting for 79.1% of the total Internet users (http://www.gov.cn/xinwen/2021-02/03/content5584518.html (accessed on 3 February 2021)). As the proportion of online consumption continues to increase, the selling modes of e-commerce platforms also show a trend of diversification. At present, there are mainly reselling modes (such as self-owned stores of Amazon.cn, Jingdong.com, and Suning.com) and agency modes (such as Tmall.com, Jingdong.com flagship store, and Suning.com flagship store) [1,2]. In addition, manufacturers (such as Apple, Dell, and Hewlett & Packard) can also sell products through self-established online channels (direct selling model) [3,4,5]. Therefore, manufacturers can sell products to consumers through two different dual-channel selling models: one is a dual-channel selling model composed of an online direct selling channel and reselling channel (direct selling channel + reselling model); the other is the dual-channel selling model (direct selling channel + agency model) composed of online direct selling channels and agency selling channels. Therefore, the appropriate choice of dual-channel selling modes by manufacturers is an important issue discussed in this paper. 

At the same time, logistics service plays a very important role in the process of product selling, which will affect consumer demand satisfaction (Leng and Parlar [6]; Yu et al. [7]). About 93% of consumers believe that logistics service is an important factor affecting their overall shopping experience (Boyaci and Ray [8]; Pekgun et al. [9]). Many e-commerce platform enterprises have realized the importance of logistics services, through the establishment of their own logistics systems, and strive to provide consumers with perfect logistics services. For example, Amazon.cn has invested a large amount of money in the construction of air cargo centers and the improvement of its transportation fleet so as to further improve its logistics service level. Jingdong.com is known for its fast and accurate delivery service. Manufacturers, on the one hand, can choose the logistics services provided by third-party logistics service providers. On the other hand, logistics services provided by e-commerce platforms can also be selected. Because logistics service sharing plays a very important role in reducing energy consumption, protecting the natural environment, and increasing corporate profits, it has attracted wide attention (He et al. [10]). For example, Jingdong.com, one of China’s largest e-commerce platforms, has been sharing its superior logistics services with merchants on its platform, such as online bookstores, since 2016. If there is no shared logistics service provided by Jingdong.com, merchants can only choose logistics services provided by third-party logistics service providers (such as ZTO Express and STO Express). Therefore, the appropriate choice of logistics service mode by manufacturers is another important issue discussed in this paper.

Motivated by the above analysis, this study attempts to answer the following research questions:(1)What kind of dual-channel selling modes should manufacturers choose in this paper in the green low-carbon supply chain?(2)What kind of logistics service modes should manufacturers choose in this paper in the green low-carbon supply chain?

As is known to us, larger numbers of scholars have discussed the logistics service or the selling channel selection strategy in traditional supply chains. Some studies have discussed operation management in green supply chain management. However, few studies analyze the logistics service strategy and the selling channel selection strategy in a green low-carbon supply chain which includes one manufacturer and one e-commerce platform. To fill this gap, a green low-carbon supply chain is selected as a research object for the first time. Moreover, we introduce the logistics service strategy and the selling channel selection strategy in the green low-carbon supply chain.

Thus, we consider a green low-carbon supply chain consisting of one manufacturer and one e-commerce platform. The manufacturer can choose two selling modes to sell products: one is the dual-channel selling mode consisting of the direct selling channel and reselling channel; the other is the dual-channel selling mode composed of the online direct selling channel and agency channel. At the same time, the manufacturer can choose two logistics service modes to distribute products: one is to choose the logistics service provided by third-party logistics service providers (such as SF Express and Yunda Express); the other is to choose logistics services provided by e-commerce platforms (such as Jingdong.com Logistics). In the process of selling and distributing products, two important decisions need to be made: Which dual-channel selling modes should the manufacturer choose to sell the product? In different dual-channel selling modes, which logistics service mode should manufacturers choose to distribute products?

The following results were obtained: In the direct selling channel and reselling channel, when the basic market demand and the logistics service level of the third-party logistics service provider are low, manufacturers will choose the e-commerce platform logistics service; in the opposite case, manufacturers will choose the third-party logistics service. In the direct selling channel and agency channels, when the logistics service level of the third-party logistics service provider is greater than or equal to a certain critical value and less than or equal to the logistics service level of the e-commerce platform, manufacturers will choose the e-commerce platform logistics service; in the opposite case, manufacturers will choose the third-party logistics service. No matter whether the manufacturer chooses the logistics service provided by the third-party logistics service provider or the logistics service provided by the e-commerce platform, the manufacturer should choose the direct selling channel and the agency channel.

This study contributes to the literature by considering the optimal decision of a green low-carbon supply chain. This study brings together the literature from streams on the selling channel selection strategy in green supply chains and the logistics service strategy in green supply chains. The impacts of the logistics service cost, the selling cost, and the green input cost coefficient on the optimal decision and the firms’ profit are discussed.

The remainder of this paper is organized as follows: Section 2 provides the literature review. Section 3 builds and solves the model. The selection of the logistics service mode under the direct selling channel + reselling channel mode is analyzed in Section 4. The selection of the logistics service mode under the direct selling channel and agency channel mode is analyzed in Section 5. The numerical analysis is presented in Section 6. Finally, the conclusions are presented in Section 7.

## 2. Literature Review

In recent years, problems related to enterprise sales models and logistics service models have attracted extensive attention in the academic circle. Many scholars have engaged in in-depth discussions on the factors affecting the selection of sales models and pricing decisions of manufacturers. For example, Jiang et al. [11] analyzed how manufacturers should choose sales models when faced with uncertain demand. With the continuous improvement of the efficiency of online sales, Liu et al. [12] pointed out that manufacturers prefer to sell products in resale mode. Dan et al. [13] analyzed the optimal pricing decision in two dual-channel sales models, centralized decision making and decentralized decision making. Tan and Yi [14] analyzed the influence of online direct delivery on the optimal pricing decision of manufacturers in s dual-channel sales model by constructing the pricing model of a distributed dual-channel supply chain and the delivery time strategy model of online direct sales under fairness concerns. The above literature mainly discusses the selection factors and pricing strategies of enterprise sales mode but does not discuss the selection of logistics service mode.

Outsourcing and sharing are two kinds of logistics service modes that have been widely used and effective in practice. Choi et al. [15] pointed out that logistics service outsourcing has become a key issue in service supply chain management. Tsai et al. [16] analyzed the potential risks leading to the failure of logistics service outsourcing. Lou et al. [17] discussed the selection of logistics service outsourcing strategies in the pure offline sales channels dominated by retailers. At the same time, many scholars have discussed the logistics service sharing problem. For example, He et al. [18] analyzed the logistics service sharing problem in a dual-channel supply chain consisting of a single manufacturer and a single retailer. Qin et al. [19] discussed the economic benefits brought by logistics service sharing to e-commerce platforms. Li et al. [20] analyzed the influence of competition intensity and logistics service sensitivity on enterprise decision making and profit. Li et al. [21] analyzed and discussed the selection of retailers’ business models, pointing out that retailers can choose a platform sales model, a distribution model, or a mixed sales model to sell products. Sun et al. [22] analyzed the influence of direct selling cost and platform transaction cost on manufacturers’ sales mode selection. Sun et al. [23] discussed the pricing strategies of traditional retailers and dual-channel retailers.

Based on the supply chain of complementary products sold by two manufacturers through different e-commerce platforms in resale or consignment mode, Tian et al. [24] studied the manufacturers’ sales mode selection and Taiwan’s logistics service decision under the circumstances of platform providers providing and not providing logistics services. Wu and Xu [25] clarified the logistics equilibrium theory and the relationship between it and the C2M mode service manufacturing logistics system, put forward the ideas and methods for the construction of a logistics equilibrium model, and proposed that the optimization goal was to maximize benefits, and the four aspects of logistics equilibrium were taken as constraints to construct and solve the equilibrium model. Dai et al. [26] used the co-agent theory to build an incentive mechanism model under the two situations of cooperation and non-cooperation between rural and urban logistics demanders and carried out a characteristic analysis and simulation experiment to determine the optimal incentive mechanism based on the solution results of the model.

This research also complements the literature on coopetition in an inter-organizational relationship. Coopetition refers to the interdependence when competition and cooperation occur simultaneously between two or more firms [27]. Many studies adopt an analytical framework to explore coopetition in a variety of contexts. For example, Anupindi et al. [28] consider the non-cooperative inventory decisions and the subsequent cooperative inventory transshipment among multiple competing retailers. Brandenburger and Stuart [29] propose a hybrid non-cooperative and cooperative game model for business coopetition strategy. Hu et al. [30] study the cooperative negotiation of revenue sharing between competing airlines.

Based on above analysis, this paper considers an e-commerce supply chain consisting of a single manufacturer and a single e-commerce platform. First of all, in the dual-channel sales mode which consists of an online direct selling channel and an online reseller channel, the logistics service mode that manufacturers should choose is discussed. Furthermore, in the dual-channel sales mode composed of an online direct selling channel and an online agency channel, the logistics service mode that manufacturers should choose is discussed. Finally, the selection of the sales model of manufacturers is analyzed. Table 1 presents the different between our paper and the previous research.

## 3. The Model

In this section, first, we introduce the theoretical model built in this paper. Second, we give the demand and relevant parameters in this paper.

### 3.1. Model Setup

This paper considers a market composed of a single manufacturer (such as Midea, Haier) and a single e-commerce platform (such as Amazon.cn, Jingdong.com). Manufacturers can choose two selling modes to sell products: one is the dual-channel selling mode consisting of the direct selling channel and reselling channel; the other is the dual-channel selling mode, which is composed of the online direct selling channel and the agency channel. At the same time, manufacturers can choose two logistics service modes to distribute products: one is to choose the logistics service provided by third-party logistics service providers (such as SF Express and Yunda Express); the other is to choose logistics services provided by e-commerce platforms (such as Jingdong.com Logistics). In the process of selling and distributing products, two important decisions need to be made: Which dual-channel selling modes should the manufacturer choose to sell the product? In different dual-channel selling modes, which logistics service mode should manufacturers choose to distribute products? Therefore, the two dual-channel selling modes and logistics service modes are shown in Figure 1. The four theoretical models are explained as follows:

### 3.2. Demand and Parameters

According to the research focus of this paper, the following basic assumptions are made in the model:

Assumption 1: According to Refs. [19,20], we assume that the product market demand is a linear function of selling price and logistics service level. (1) The market demand of an online direct selling channel is inversely proportional to its retail price and is directly proportional to the retail price of the online agency (reselling) channel. At the same time, it is directly proportional to the service level of the logistics service system adopted by it and inversely proportional to the service level of the logistics service system adopted by the agency (reselling) channel. (2) The market demand of the online agency (reselling) channel is inversely proportional to its retail price and is directly proportional to the retail price of the online direct selling channel. At the same time, it is directly proportional to the service level of the logistics service system adopted by the direct selling channel and inversely proportional to the service level of the logistics service system adopted by the direct selling channel. Therefore, the demand for online direct selling channels in ND and NC models is $D_{m}=\theta -p_{2}+\alpha p_{1}+\beta L_{2}-\gamma L_{1}$, and the demand for the reselling (agency) channel is $D_{e}=\theta -p_{1}+\alpha p_{2}+\gamma L_{1}-\beta L_{2}$; the demand for online direct selling channels in SD and SC models is $D_{m}=\theta -p_{2}+\alpha p_{1}+\beta L_{s}-\gamma L_{s}$, and the demand for the reselling (agency) channel is $D_{e}=\theta -p_{1}+\alpha p_{2}+\gamma L_{s}-\beta L_{s}$.

Assumption 2: With reference to Refs. [20,21], this paper considers that manufacturers only sell the same product in the direct selling channel or the reselling (agency) channel and that the basic market demand in the direct selling channel and the reselling (agency) channel is the same (without considering the impact of market share differentiation).

Assumption 3: Referring to [29], in the dual-channel sales model of direct selling channel + reselling channel, there is a certain quantitative relationship between the retail price $p_{1}$ set by the reselling channel and the wholesale price w paid to the manufacturer by the e-commerce platform; that is, $p_{1}=w+u$, where u is the premium of each unit product sold by the e-commerce platform in the reselling channel.

Assumption 4: Referring to [31,32], it is assumed that the marginal cost of products is normalized to zero, and both manufacturers and e-commerce platforms have enough inventory to meet customer demand.

Assumption 5: With reference to [33], it is assumed that each delivery of the e-commerce platform will generate variable logistics service costs c (including delivery for itself and delivery for manufacturers). Meanwhile, in order to simplify the calculation, one-time input costs of self-established logistics of the e-commerce platform are not considered.

Assumption 6: When a manufacturer sells a new product through the consignment model, it is assumed that the commission rate ρ charged by the network platform is an exogenous variable and satisfies 0≤ρ≤1. Similar assumptions can be seen in the work of Zhang et al. [34], and Ha et al. [35]. This is because online platforms (such as Amazon.cn and Jingdong.com) usually charge the same commission rates for many categories of products, and these commission rates can be directly queried through the official websites of online platforms.

Assumption 7: Similar to the studies of Ha et al. [35], Ren et al. [36], Qin et al. [19], and Lai et al. [37], it is assumed that a Stackelberg game relationship is formed between the manufacturer and the network platform, with the manufacturer in a dominant position and the network platform in a subordinate position. In addition, both manufacturers and network platforms are perfectly rational and risk-neutral, with symmetrical information and profit maximization.

The symbols and descriptions involved in the model are shown in Table 2.

## 4. Selection of Logistics Service Mode under Direct Selling Channel + Reselling Channel Mode

In this section, first, we build a third-party logistics service model, and then the logistics service model of an e-commerce platform is built. Last, we compare the results.

### 4.1. Third-Party Logistics Service Model (ND Model)

In this model, the manufacturer and the e-commerce platform follow a Stackelberg game: In the first stage, the manufacturer first decides the wholesale price $w^{ND}$ and the retail price of the direct selling channel $p_{2}^{ND}$ under the given logistics service levels $L_{1}$ and $L_{2}$. In the second stage, the e-commerce platform decides the retail price $p_{1}^{ND}$. The manufacturer profit function is as follows:(1)$$\Pi_{m}{=\mathrm{wD}}_{e}+\left(p_{2}-p_{m}\right)D_{m}$$

The profit function of the e-commerce platform is as follows:(2)$$\Pi_{e}=\left(p_{1}-w-c\right)D_{e}$$

By using backward induction, the equilibrium solutions of all items in this model can be obtained as follows:$$w^{ND*}=\frac{\theta -c+\theta \alpha +c\alpha^{2}+\gamma L_{1}-\alpha \gamma L_{1}-\beta L_{2}+\alpha \beta L_{2}}{2\left(1-\alpha^{2}\right)}$$
$$p_{1}^{ND*}=\frac{\left(3-2\alpha -\alpha^{2}\right)\left(\gamma L_{1}-\beta L_{2}\right)+\left(1+\alpha \right)\left(\theta \left(3-\alpha \right)+\left(1-\alpha \right)\left(c+\alpha p_{m}\right)\right)}{4\left(1-\alpha^{2}\right)}$$
$$p_{2}^{ND*}=\frac{\theta \left(1+\alpha \right)+\left(\beta L_{2}-\gamma L_{1}\right)\left(1-\alpha \right)+p_{m}\left(1-\alpha^{2}\right)}{2\left(1-\alpha^{2}\right)}$$

### 4.2. Logistics Service Model of E-Commerce Platform (SD Mode)

In this model, the manufacturer and the e-commerce platform follow a Stackelberg game: In the first stage, under the premise of the logistics service level $L_{s}$ of the e-commerce platform, the manufacturer first decides the wholesale price $w^{SD}$ and retail price $p_{2}^{SD}$ of the direct sales channel. In the second stage, the e-commerce platform decides the retail price $p_{1}^{SD}$. The manufacturer profit function is as follows:(3)$$\Pi_{m}{=\mathrm{wD}}_{e}+\left(p_{2}-p_{e}\right)D_{m}$$

The profit function of the e-commerce platform is as follows:(4)$$\Pi_{e}=\left(p_{1}-w-c\right)D_{e}+\left(p_{e}-c\right)D_{m}$$

By using backward induction, the equilibrium solutions of all items in this model can be obtained as follows:$$w^{SD*}=\frac{\theta +\theta \alpha +c\left(\alpha^{2}-1\right)\left(1-\alpha \right)-L_{s}\left(\beta -\gamma \right)\left(1-\alpha \right)-\alpha p_{e}+\alpha^{3}p_{e}}{2\left(1-\alpha^{2}\right)}$$
$$p_{1}^{SD*}=\frac{{\left(-1+\alpha \right)}^{2}\left(\beta -\gamma \right)L_{s}-\left(1+\alpha \right)\left(3\theta +c-\theta \alpha -2c\alpha +c\alpha^{2}-2\left(-1+\alpha \right)\alpha p_{e}+2\left(-1+\alpha \right)\beta L_{s}+2\gamma L_{s}-2\alpha \gamma L_{s}\right)}{4\left(-1+\alpha^{2}\right)}$$
$$p_{2}^{SD*}=\frac{\theta \left(1+\alpha \right)+L_{s}\left(\beta -\gamma \right)\left(1-\alpha \right)+p_{e}\left(1-\alpha^{2}\right)}{2\left(1-\alpha^{2}\right)}$$

### 4.3. Comparative Analysis

**Theorem 1.** 
*In the dual-channel sales model of direct selling channel + reselling channel, by comparing the retail prices of the third-party logistics service model and the e-commerce platform logistics service model, we can obtain the following:*
(1)
*when*

${\bar{L}}_{2}^{ND}\leq L_{2}^{ND}\leq L_{s}^{SD}$

*,*

$p_{1}^{SD*}>p_{1}^{ND*}$

*;*
(2)*when*${\bar{L}}_{2}^{ND}\leq L_{2}^{ND}\leq L_{s}^{SD}$*,*$p_{2}^{SD*}>p_{2}^{ND*}$.


Theorem 1 shows the following: (1) When the logistics service level of the third-party logistics service provider is greater than or equal to a certain critical value and less than or equal to the logistics service level of the e-commerce platform, the retail price of the reselling channel under the logistics service mode of the e-commerce platform is greater than the retail price of the reselling channel under the third-party logistics service mode. The main reason for this phenomenon is that when the logistics service level of the e-commerce platform is higher than that of the third-party logistics service provider, the e-commerce platform can charge the manufacturer a higher price for logistics service. This is because the logistics service price and reselling channel retail price are positively correlated. Therefore, the higher the logistics service price charged by the e-commerce platform is, the higher the retail price of the reselling channel will be. (2) When the logistics service level of the third-party logistics service provider is greater than or equal to a certain critical value and less than or equal to the logistics service level of the e-commerce platform, the retail price of the direct selling channel under the logistics service mode of the e-commerce platform is also greater than the retail price of the direct selling channel under the third-party logistics service mode. The main reasons are as follows: When the logistics service level of the third-party logistics service provider is low, the manufacturer will choose the logistics service provided by the e-commerce platform, and then the e-commerce platform will take the opportunity to charge the manufacturer a higher price for logistics service. Since the price of logistics service is also positively correlated with the retail price of direct selling channels, manufacturers can only maintain their profit level through a high price strategy.

**Theorem 2.** 
*In the dual-channel selling model of direct selling channel + reselling channel, by comparing the profits of manufacturers in the third-party logistics service model and the e-commerce platform logistics service model, we can obtain the following:*
(1)
*when*

$\theta \leq \tilde{\theta }$

*,*

$L_{2}^{ND}\leq {\tilde{L}}_{2}^{ND}$

*,*

$\Pi_{m}^{SD*}\geq \Pi_{m}^{ND*}$

*;*
(2)*when*$\theta >\tilde{\theta }$*,*$L_{2}^{ND}>{\tilde{L}}_{2}^{ND}$*,*$\Pi_{m}^{SD*}<\Pi_{m}^{ND*}$.


Theorem 2 shows the following: (1) When the basic market demand and the logistics service level of the third-party logistics service provider are low, the manufacturer will choose the logistics service mode of the e-commerce platform rather than the third-party logistics service mode. At this time, the logistics service model of the e-commerce platform is very beneficial to manufacturers. (2) When the basic market demand and the logistics service level of the third-party logistics service provider are high, manufacturers will choose the third-party logistics service mode rather than the e-commerce platform logistics service mode. When the manufacturer chooses the third-party logistics service mode, the manufacturer has the dominant right of pricing, which makes the third-party logistics service providers strive to improve their logistics service level and provide better logistics services for the manufacturer. Therefore, when the logistics service level of the third-party logistics service provider is higher, it is more advantageous for the manufacturer to choose the third-party logistics service mode.

**Theorem 3.** 
*In the dual-channel selling model of direct selling channel + reselling channel, the profit of the e-commerce platform can be obtained by comparing the third-party logistics service model and the e-commerce platform logistics service model.*
(1)
*when*

$\theta \leq \hat{\theta }$

*,*

$L_{2}^{ND}\leq {\hat{L}}_{2}^{ND}$

*,*

$\Pi_{e}^{SD*}\leq \Pi_{e}^{ND*}$

*;*
(2)*when*$\theta >\hat{\theta }$*,*$L_{2}^{ND}>{\hat{L}}_{2}^{ND}$*,*$\Pi_{e}^{SD*}>\Pi_{e}^{ND*}$.


Theorem 3 shows the following: (1) When the basic market demand and the logistics service level of the third-party logistics service provider are both low, the e-commerce platform is unwilling to provide logistics services to the manufacturer. At this time, if the e-commerce platform provides logistics services to the manufacturer, its profit will be damaged. (2) When the basic market demand and the logistics service level of the third-party logistics service provider are high, the e-commerce platform is willing to provide logistics services to the manufacturer. At this time, the e-commerce platform can earn more profit by providing logistics services to the manufacturer.

## 5. Selection of Logistics Service Mode under the Mode of Direct Selling Channel and Agency Channel

In this section, first, we build a third-party logistics service model, and then the logistics service model of an e-commerce platform is built. Last, we compare the results.

### 5.1. Third-Party Logistics Service Model (NC Model)

In this model, given the logistics service levels $L_{1}$ and $L_{2}$, the manufacturer directly decides the retail price of the direct selling channel $p_{2}^{NC}$ and the commission channel $p_{1}^{NC}$. The manufacturer profit function is as follows:(5)$$\Pi_{m}=\left(1-\rho \right)p_{1}D_{e}+\left(p_{2}-p_{m}\right)D_{m}$$

The profit function of the e-commerce platform is as follows:(6)$$\Pi_{e}=\left(\rho p_{1}-c\right)D_{e}$$

All equilibrium solutions in this model can be obtained by solving the following:$$p_{1}^{NC*}=\frac{-2\left(1+\alpha \right)\theta +L_{1}\gamma \left(-\alpha^{2}\left(\rho -2\right)+2\left(\rho -1\right)\right)+2\theta \rho +\alpha \rho \left(p_{m}-L_{2}\beta +\theta \right)}{\alpha^{2}{\left(\rho -2\right)}^{2}+4\left(1-\rho \right)}$$
$$p_{2}^{NC*}=\frac{p_{m}\left(2\left(\rho -1\right)-\alpha^{2}\left(\rho -2\right)\right)+\left(\rho -1\right)\left(2\theta \left(1+\alpha \right)+L_{2}\beta \left(2+\alpha^{2}\left(\rho -2\right)\right)-\alpha \rho \left(L_{1}\gamma +\theta \right)\right)}{\alpha^{2}{\left(\rho -2\right)}^{2}+4\left(1-\rho \right)}$$

### 5.2. Logistics Service Model of E-Commerce Platform (SC Model)

In this model, under the premise of the given logistics service level of the e-commerce platform $L_{s}$, manufacturers directly decide the retail price of the direct selling channel $p_{2}^{SC}$ and agency channel $p_{1}^{SC}$. The manufacturer profit function is as follows:(7)$$\Pi_{m}=\left(1-\rho \right)p_{1}D_{e}+\left(p_{2}-p_{e}\right)D_{m}$$

The profit function of the e-commerce platform is as follows:(8)$$\Pi_{e}=\left(\rho p_{1}-c\right)D_{e}+\left(p_{e}-c\right)D_{m}$$
$$p_{1}^{SC*}=\frac{\theta \left(1+\alpha \right)\left(1-\rho \right)+\alpha^{2}\rho +\gamma L_{s}\left(1+\alpha \right)\left(1-\alpha -\rho \right)}{2\left(1-\alpha^{2}\right)\left(1-\rho \right)}$$
$$p_{2}^{SC*}=\frac{\begin{array}{c} \alpha \left(-2+L_{s}\gamma +\theta \right)\left(1-\rho \right)+2\left(L_{s}\beta +L_{s}\gamma +\theta \right)\left(1-\rho \right)+\alpha^{3}\left(2-L_{s}\gamma -\rho \right)\\ -\alpha^{2}\left(2L_{s}\beta +2L_{s}\gamma +\theta -\rho \left(2L_{s}\beta +L_{s}\gamma +\theta \right)\right) \end{array}}{2\left(1-\alpha^{2}\right)\left(1-\rho \right)}$$

### 5.3. Manufacturer Logistics Service Model Selection Strategy

This section mainly analyzes the influence of the basic market demand and the logistics service level of the third-party logistics service provider on the selection strategy of the manufacturer’s logistics service mode in the dual-channel selling mode of a direct selling channel and an agency channel.

**Theorem 4.** 
*In the dual-channel selling model of a direct selling channel and an agency channel, by comparing the retail prices of the third-party logistics service model and the e-commerce platform logistics service model, we can obtain the following:*
(1)
*when*

L¯2NC*≤L2NC≤LsSC

*,*

$p_{1}^{SC*}>p_{1}^{NC*}$

*;*
(2)*when*${\bar{L}}_{2}^{NC*}\leq L_{2}^{NC}\leq L_{s}^{SC}$*,*$p_{2}^{SC*}>p_{2}^{NC*}$.


Theorem 4 shows the following: (1) When the logistics service level of the third-party logistics service provider is greater than or equal to a certain critical value and less than or equal to the logistics service level of the e-commerce platform, the retail price of the agency channel under the logistics service mode of the e-commerce platform is greater than the retail price of the agency channel under the third-party logistics service mode. The reasons for this phenomenon are as follows: When the logistics service level of the e-commerce platform is higher than that of the third-party logistics service provider, the e-commerce platform can charge a higher logistics service price to the manufacturer to ensure its own profit level. In order to make up for the profit loss caused by the excessively high logistics service price, the manufacturer will set a relatively high retail price. (2) When the logistics service level of the third-party logistics service provider is greater than or equal to a certain critical value and less than or equal to the logistics service level of the e-commerce platform, the retail price of the direct selling channel under the logistics service mode of the e-commerce platform is also greater than the retail price of the direct selling channel under the third-party logistics service mode. The main reasons are as follows: When the logistics service level of the third-party logistics service provider is low, the manufacturer will also choose the logistics service provided by the e-commerce platform, and then the e-commerce platform will take the opportunity to charge the manufacturer a higher price for logistics service. The manufacturer can only maintain its profit level through the high price strategy. The manufacturer will also set a relatively high retail price.

**Theorem 5.** 
*In the dual-channel sales model of a direct selling channel and an agency channel, by comparing the profits of manufacturers in the third-party logistics service model and the e-commerce platform logistics service model, we can obtain the following:*
(1)
*when*

L¯2NC*≤L2NC≤LsSC

*,*

$\Pi_{m}^{SC*}\geq \Pi_{m}^{NC*}$

*;*
(2)*when*${\bar{L}}_{2}^{NC*}>L_{2}^{NC}>L_{s}^{SC}$*,*$\Pi_{m}^{SC*}<\Pi_{m}^{NC*}$.


Theorem 5 shows the following: (1) When the logistics service level of the third-party logistics service provider is greater than or equal to a certain critical value and less than or equal to the logistics service level of the e-commerce platform, the profit of the manufacturer under the logistics service mode of the e-commerce platform is greater than or equal to the profit of the manufacturer under the third-party logistics service mode. In other words, manufacturers will choose the logistics service model of the e-commerce platform rather than the third-party logistics service model. (2) When the logistics service level of the third-party logistics service provider is greater than that of the e-commerce platform, the profit of the manufacturer under the logistics service mode of the e-commerce platform is smaller than that of the manufacturer under the third-party logistics service mode. That is to say, manufacturers will choose the third-party logistics service model rather than the e-commerce platform logistics service model. The main reasons for this phenomenon are as follows: In the dual-channel selling model of direct selling channel + agency channel, the e-commerce platform itself does not sell products, but only obtains certain profits by charging a certain percentage of commission through the commission channels, without any dominant pricing power. Relatively speaking, manufacturers have the dominant pricing power in both direct and agency channels. With the changes in the basic potential market demand and the logistics service level of third-party logistics service providers, manufacturers can adjust the retail price of products independently. Therefore, the profit obtained by the manufacturer will change with the basic market demand and the logistics service level of the third-party logistics service provider.

**Theorem 6.** 
*In the dual-channel sales model of a direct selling channel and an agency channel, by comparing the profits of the e-commerce platform in the third-party logistics service model and the e-commerce platform logistics service model, we can obtain the following:*
(1)
*when*

${\bar{L}}_{2}^{NC**}\leq L_{2}^{NC}\leq L_{s}^{SC}$

*,*

$\Pi_{e}^{SC*}\geq \Pi_{e}^{NC*}$

*;*
(2)*when*${\bar{L}}_{2}^{NC**}>L_{2}^{NC}>L_{s}^{SC}$*,*$\Pi_{e}^{SC*}<\Pi_{e}^{NC*}$.


Theorem 6 shows the following: (1) When the logistics service level of the third-party logistics service provider is greater than or equal to a certain critical value and less than or equal to the logistics service level of the e-commerce platform, the profit obtained by the e-commerce platform when it provides logistics service to the manufacturer is greater than the profit obtained by the e-commerce platform under the third-party logistics service mode. Obviously, when the logistics service level of the third-party logistics service provider is lower than that of the e-commerce platform, the manufacturer is willing to choose the logistics service provided by the e-commerce platform. (2) When the logistics service level of the third-party logistics service provider is greater than that of the e-commerce platform, the profit obtained by the e-commerce platform when it provides logistics service to the manufacturer is smaller than that obtained by the e-commerce platform under the third-party logistics service mode.

### 5.4. Sales Mode Selection

This section mainly analyzes whether manufacturers should choose the direct selling channel + reselling channel mode or the direct selling channel + agency channel mode for product sales under the third-party logistics service mode and e-commerce platform logistics service mode.

**Theorem 7.** 
*In the third-party logistics service mode, by comparing the profit of the manufacturer in the NC mode and the ND mode, we can obtain the following:*
(1)
*when*

$L_{2}^{**}\leq L_{2}$

*,*

ΠmNC*≥ΠmND*

*;*
(2)*when*$L_{2}^{**}>L_{2}$*,*$\Pi_{m}^{NC*}<\Pi_{m}^{ND*}$.


Theorem 7 shows the following: When the logistics service level of the third-party logistics service provider is greater than or equal to a certain critical value, under the third-party logistics service mode, the profit of the manufacturer under the direct selling channel + agency channel dual-channel selling mode is greater than the profit of the manufacturer under the direct selling channel + reselling channel dual-channel selling mode. In other words, when the manufacturer chooses the logistics service provided by the third-party logistics service provider, it should choose the dual-channel selling model of direct selling and agency channels to sell products.

**Theorem 8.** 
*In the logistics service model of an e-commerce platform, by comparing the profit of the manufacturer in SC mode and SD mode, we can obtain the following:*
(1)
*when*

$L_{2}\leq L_{s}$

*,*

ΠmNC*≥ΠmND*

*;*
(2)*when*$L_{2}>L_{s}$*,*$\Pi_{m}^{NC*}<\Pi_{m}^{ND*}$.


Theorem 8 shows that when the logistics service level of an e-commerce platform is greater than that of third-party logistics service providers, manufacturers will choose the logistics service provided by the e-commerce platform. In this case, the manufacturer’s profit under the dual-channel selling model of direct selling and agency channels is still greater than the manufacturer’s profit under the dual-channel selling model of direct selling and reselling channels. At this time, manufacturers should still choose the dual-channel sales model of direct selling and agency channels to sell products.

**Theorem 9.** 
*(i) If*

$L_{2}^{N}\leq {\hat{L}}_{2}^{N}$

*and*

$\theta \leq \hat{\theta }$

*, the e-commerce platform will not share its logistics service with the manufacturer; in other words, the equilibrium mode is no service-sharing;*
*(ii)* 
*If*

$L_{2}^{N}>{\hat{L}}_{2}^{N}$

*and*

$\theta >\hat{\theta }$

*, the e-commerce platform share would like to share its logistics service and the manufacturer will accept the e-commerce platform’s provision of logistics service sharing; in other words, the equilibrium mode is service sharing.*



Theorem 9 shows that the 3PL’s logistics service level and the magnitude of the market potential will jointly alter the equilibrium mode. In other words, when the 3PL’s logistics service level and the market potential are relatively low, the equilibrium mode is no service-sharing. The reason is that in such regions the e-commerce platform will be worse off from logistics service sharing; hence, the e-commerce platform will not share its logistics service with the manufacturer. However, as the 3PL’s logistics service level, the market potential, or both increase, the equilibrium mode will evolve from no service-sharing to service sharing. The main reason is that the e-commerce platform will benefit from logistics service sharing in such regions. However, when the 3PL’s logistics service level or the market potential is very high, the manufacturer will not accept the e-commerce platform’s offer of logistics service sharing.

## 6. Numerical Analysis

By comparing the profit function of manufacturers in different logistics service modes and selling modes, this paper preliminarily explains which logistics service mode and selling mode manufacturers should choose. This section further explains whether the logistics service mode of an e-commerce platform is beneficial to both manufacturers and e-commerce platforms using a numerical simulation method. Due to the analytical tractability, we mainly rely on numerical studies to obtain the results. Without loss of generality, the basic parameters are set as follows: α=0.3, β=0.8, γ=0.4, ρ=0.04, $L_{1}=L_{s}=8$, c=0.6, $p_{m}=1$. The numerical simulation results are shown in Figure 2 and Figure 3.

In the dual-channel selling model of direct selling channel + reselling channel, when the logistics service level and basic market demand of third-party logistics service providers are at a low level (region), the logistics service model of the e-commerce platform is favorable to the manufacturer but unfavorable to the e-commerce platform. When the logistics service level and basic market demand of the third-party logistics service provider are at a high level (region), the logistics service mode of an e-commerce platform is favorable to the e-commerce platform but unfavorable to the manufacturer. In the above two regions, the impact of e-commerce platform logistics service mode on manufacturers and e-commerce platforms is completely opposite. However, when the logistics service level of the third-party logistics service provider and the basic market demand are at the medium level (region), both the manufacturer and the e-commerce platform can benefit from the logistics service mode of the e-commerce platform.

As shown in Figure 3, when the logistics service level of the third-party logistics service provider is greater than a certain critical value, the logistics service mode of the e-commerce platform is beneficial to both the manufacturer and the e-commerce platform, both of which are in a “win–win” state (region). When the logistics service level of the third-party logistics service provider is less than a certain critical value, the logistics service mode of the e-commerce platform is unfavorable to both the manufacturer and the e-commerce platform, both of which are in a “lose–lose” state (region).

This study also finds a more interesting phenomenon: Different from the dual-channel selling model of a direct selling channel and an agency channel, there is no certain region in the dual-channel selling model of a direct selling channel and a reselling channel in which the logistics service model of an e-commerce platform can cause losses to both manufacturers and e-commerce platforms. However, there is a region in the dual-channel sales model of a direct selling channel and an agency channel in which the logistics service model of an e-commerce platform can make both manufacturers and e-commerce platforms suffer losses.

## 7. Conclusions

In this paper, firstly, in a green low-carbon supply chain composed of manufacturers and e-commerce platforms, the selection of logistics service mode in the direct selling + reselling dual-channel selling model is analyzed. Secondly, the paper analyzes the choice of logistics service mode in the dual-channel selling mode of direct selling and agency channels. Finally, the selection of the selling mode of manufacturers is analyzed. By comparing the optimal decision under different logistics service modes and selling models, we reach many innovative conclusions.

The results show the following: (1) In different logistics service modes, with the continuous increase in the basic potential demand of the market, the profits of manufacturers and e-commerce platforms are also constantly improving. (2) In the dual-channel selling model of direct selling + agency channel, in the logistics service model of an e-commerce platform, the retail price of the direct selling channel and agency channel is higher than that of the third-party logistics service model. The same conclusion can be reached in the dual-channel selling model of direct selling and reselling channels. (3) In the dual-channel sales model of direct selling and agency channels, in the logistics service model of an e-commerce platform, the profit of the manufacturer and the profit of the e-commerce platform are greater than those of the third-party logistics service model. In the direct selling + reselling dual-channel sales mode, only when the logistics service level of the third-party logistics service provider and the basic potential market demand meet some specific conditions, the profit of the manufacturer and the e-commerce platform in the logistics service mode is greater than that in the third-party logistics service mode. (4) No matter whether the manufacturer chooses the logistics service provided by the third-party logistics service provider or the e-commerce platform, the manufacturer should choose the dual-channel sales model of direct sales and commission sales to sell its products.

Our findings provide executable managerial insights for the platforms and manufacturers in the e-commerce market. For example, a platform should share its logistics service with the manufacturer when the 3PL’s logistics service level is not low (i.e., when the platform does not have much competitive edge in logistics service). Moreover, it is beneficial for the platform to share its logistics service for popular product categories (i.e., product categories with high market potential). Furthermore, if the manufacturer would not like to engage in logistics service sharing, the platform can provide certain preferential treatments.

This paper can be further studied from the following aspects: (1) This paper does not consider that consumers need to pay additional logistics costs to e-commerce platforms or third-party logistics service providers. One of the future research directions is that e-commerce platforms or third-party logistics service providers should consider charging additional logistics costs to consumers. (2) This paper does not consider including the third-party logistics service provider in the game process. In the future, the third-party logistics service provider can be considered as an independent decision-making body to participate in the game process; it is hoped that this will result in more interesting conclusions being drawn. (3) This paper takes logistics service level and sales commission rate as exogenous variables. In the future, we will consider taking them as decision variables.

## Figures and Tables

**Figure 1 ijerph-20-03356-f001:**
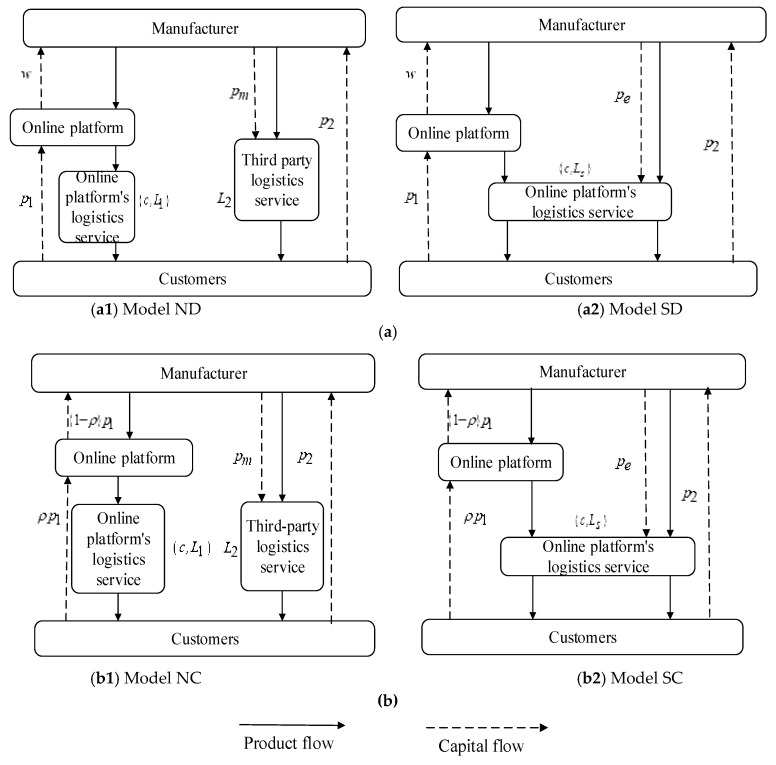
**Theoretical models of two dual-channel selling modes and logistics service modes.** (**a**) Logistics service mode under the dual-channel structure of direct selling channel and reselling channel. In the dual-channel structure of direct selling channel + reselling channel, manufacturers sell products to e-commerce platforms at wholesale prices w, and the e-commerce platforms sell products to customers at retail prices $p_{1}$, while manufacturers sell products directly to customers at retail prices $p_{2}$ through direct channels (see Figure 1a). On the one hand, the logistics service system of the e-commerce platform only provides logistics services for itself, and its logistics service level is expressed as $L_{1}$. In the direct selling channel, the manufacturer buys logistics service from the third-party logistics service provider at unit price $p_{m}$, and its logistics service level is expressed as $L_{2}$, called the third-party logistics service model (see Figure 1a(1)). On the other hand, the logistics service system of the e-commerce platform not only provides logistics services for itself but also provides logistics services for manufacturers. Manufacturers purchase logistics services from the e-commerce platform at unit price $p_{e}$ and use the e-commerce platform to represent their logistics service level $L_{s}$, which is called the logistics service mode of the e-commerce platform (see Figure 1a(2)). (**b**) Logistics service mode under the dual-channel structure of direct selling and agency channels. In the dual-channel structure of direct selling channel and agency channel, manufacturers sell products to customers through an e-commerce platform at retail price $p_{1}$, and the e-commerce platform charges a proportional commission ρ for each unit of product. Meanwhile, manufacturers sell products directly to customers at retail price $p_{2}$ through the direct selling channel (see Figure 1b). First, the logistics service system of the e-commerce platform only provides logistics services for itself, and its logistics service level is expressed as $L_{1}$. The manufacturer purchases logistics service from the third-party logistics service provider at unit price $p_{m}$, and its logistics service level is expressed as $L_{2}$, called the third-party logistics service model (see Figure 1b(1)). Second, the logistics service system of the e-commerce platform not only provides logistics services for itself but also provides logistics services for manufacturers. Manufacturers purchase logistics services from the e-commerce platform at unit price $p_{e}$ and use the e-commerce platform to represent their logistics service level $L_{s}$, which is called the logistics service model of an e-commerce platform (see Figure 1b(2)).

**Figure 2 ijerph-20-03356-f002:**
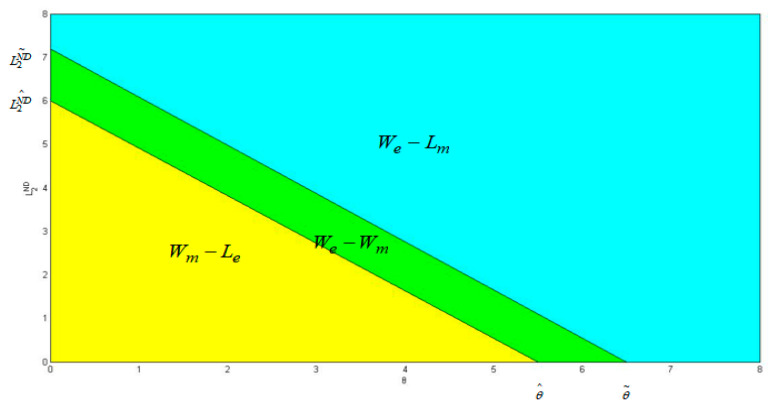
The impact of θ and $L_{2}^{ND}$ on corporate profits.

**Figure 3 ijerph-20-03356-f003:**
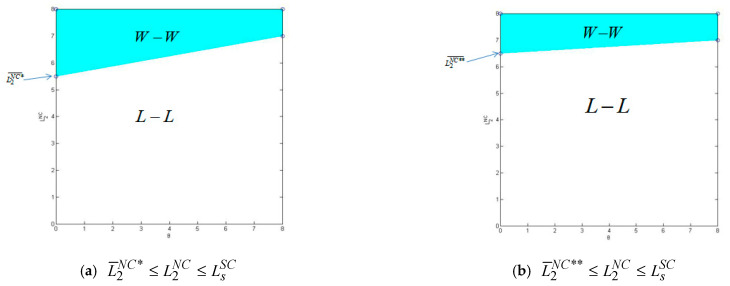
The impact of θ and $L_{2}^{ND}$ on firms’ profit.

**Table 1 ijerph-20-03356-t001:** The difference between our paper and the previous research.

Study	Forward Supply Chain	Reverse Supply Chain	Selling Channel Selection		Logistics Service Selection	
	Pricing Decision	Greenness Level Decision	Reselling Channel	Agency Channel	The 3PL Firm	The Online Platform
Liu et al. [12]	√		√			
Choi et al. [15]	√		√		√	
Tsai et al. [16]	√			√	√	
He et al. [18]	√			√		√
Qin et al. [19]	√		√		√	
Li et al. [21]	√			√		√
Sun et al. [22]	√		√			√
Tian et al. [24]	√		√		√	
Wu and Xu [25]		√			√	
Dai et al. [26]		√	√			√
Our paper	√	√	√	√	√	√

**Table 2 ijerph-20-03356-t002:** Description of the relevant notations.

Variables	Notation	Description
Decision variables	w	Wholesale prices of products obtained by e-commerce platforms through reselling channels
	$p_{1}$	Retail price of products in reselling (agency) channels
	$p_{2}$	Retail prices of products in direct selling channels
Relevant parameters	θ	Basic demand of the market
	α	Cross-price elasticity coefficient, 0<α<1
	β	The sensitivity of market demand to the logistics service level of third-party logistics service providers, β>0
	γ	The sensitivity of market demand to the logistics service level of the e-commerce platform, γ>0
	ρ	Commission rate for each unit of product sold by e-commerce platform
	c	Each delivery on an e-commerce platform generates a variable logistics service cost
	u	The premium per unit of product sold by e-commerce platforms in reselling channels
	$p_{m}$	The price of logistics service provided by the third-party logistics service provider
	$p_{e}$	The e-commerce platform-provided price of logistics service
	$L_{1}$	The logistics service level of the e-commerce platform in the third-party logistics service model
	$L_{2}$	The logistics service level of the third-party logistics service provider in the third-party logistics service model
	$L_{s}$	The logistics service level of the e-commerce platform in the logistics service model
	$D_{e}$	Demand for reselling or agency channels
	$D_{m}$	Demand for direct selling channels
	$\pi_{e}$	The e-commerce platform’s profit
	$\pi_{m}$	Manufacturer’s profit

## Data Availability

Not applicable.

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
