# Peer review of "Logistics Service Selection Strategy of Green Manufacturers in Green Low-Carbon Supply Chain"

_ijerph, 2023, doi:10.3390/ijerph20043356_

Round 1

Reviewer 1 Report

The problem under study in this manuscript can be interesting for readers. But this manuscript suffers from some major shortcomings.
1. The abstract section doesn't reflect the theoretical and practical contributions of the introduced model.
2. The organization of the introduction is acceptable. However, I think the authors should clarify the research contributions in this section. Also, the authors should place this part starting with "The following results were obtained..." after discussing the contributions.
3. I think Table 1 should be extended to demonstrate the research gap.
4. The results section should be improved by adding more analysis and discussion.
5. Comparative analysis is also needed in this manuscript.

Author Response

Dear reviewer:
Thank you for your letter and for the reviewers’ comments concerning our manuscript entitled “Logistics Service Selection Strategy of Green Manufacturers in Green Low Carbon Supply Chain” (ID: ijerph-2169812)”. Those comments are all valuable and very helpful for revising and improving our paper, as well as the important guiding significance to our research. We have studied the comments carefully and have made corrections which we hope meet with approval. Revised portions are marked in yellow in the paper. The main corrections in the paper and the responds to the reviewer’s comments are attached.

Reviewer 2 Report

The paper is well organized and it has been written with good English and use relevant and timely references. The proposed novelty is also clearly outlined. Congratulations.

The last paragraph at the end of section 1.1 Background and Research Motivation should be deleted or removed to the section 7. Conclusion. 

Minor typo is needed in Table 2 for the symbol of “The sensitivity of market demand to the logistics service level of e-commerce platform”

Add an Appendix outlining the backward induction for the equilibrium solutions in Section 4.

Add more explanation about considerations in the selection of parameters for the numerical simulations.

Minor typo for the correct title of Figure 2 and Figure 3.

In the Conclusion, please add other possible future research based on relaxing several assumptions that are used in this paper.

Author Response

(The authors gave the same response as above.)

Reviewer 3 Report

Comment 1: The topic is not new. What’s the main contribution to previous literature?

Comment 2: The logistic service level should be optimized. 

Comment 3: What's the optimal combination of sales mode and service mode? 

Comment 4: Consumer surplus should be analyzed.  

Comment 5:Alternative demand functions such as consumer utility function should be analyzed as robustness check.

Comment 6: Commision fee is exogenous. This is also a strong assumption and robustness check is needed.  

Comment 7: Writing is poor is current version. 

Author Response

(The authors gave the same response as above.)

Author Response

(The authors gave the same response as above.)

Reviewer 5 Report

Overall Review: The work reflects the dedication from the researcher. Overall the research is well explained deserves appreciation. The model section is the most impressive, which is enhancing the quality of this work. This is a very informative research. However there may be some improvements in this work for the betterment of study. The suggested comments can help the researcher to improve the quality of research paper for the mass reader. Citations must be increased throughout the research and the corresponding references to be added. English language needs a minor improvement making sentences, and paragraphs in the complete research paper, which may enhance the quality of the work.

(1)   Title

·       Authors have framed a very simple title.

·       Title needs minor modification including Analysis/ Observing/ Finding.

·       Making a technical title with more clarity would be better.

(2)   Abstract and Keywords

·       Abstract is well articulated which is representing a good snapshot of the research.

·       This section is providing a literature.

·       It would be better to add methodological information also.

·       Sentences needs a check like line no 19, 23 using semi colon then capital (;On contrary).

·       This section should be modified adding outcome and implications.

·       Keywords are well chosen and suiting with the research, but needs to check for writing style as per Applied Sciences.

·       English language needs a minor improvement

·        

(3)   Introduction

·       A good presentation of the work with the requirements.

·       Before subsection 1.1, a small paragraph can make it better.

·       Page No. 2 line No. 77, sentence needs correction (As…..).

·       Paragraphs can be made smaller.

·       Paragraphs should be evenly distributed.

·       Subsection 1.2 is too short.

·       Either subsection 1.2 can be expanded or all subsections can be deleted from Introduction.

·       Sentences are written have a proper flow and connectivity is impressive.

·       Research gap is presented with a good articulation.

·       Research questions are rightly observed.

·       Research aims are clearly presented.

·       It should provide more citations.

·       English language needs a minor improvement.

(4)   Literature review

·       Paragraphs should be evenly distributed.

·       Table 1. is very impressive and shows a collective information.

·       Some more paragraphs need to be included.

·       Current literature is too short.

·       Line no. 135-172, it can be divided in 2-3 paragraphs.

·       It should provide more citations.

·       English language needs a minor improvement.

(5)   Model

·       Before subsection 3.1, a small paragraph can make it better.

·       The research is using a good model.

·       The full form of ND, SD, NC, and SC is required.

·       There is a need of more citations.

·       It is impressive and accepted as presented.

·       Figure 1. needs a source as reference/ author’s contribution.

·       The very first paragraph needs citations.

·       Subsections are wisely chosen.

·       Subsection 3.1. must provide citations.

·       Subsection 3.2. must provide citations.

·       Assumptions are well justified.

·       Table 2. needs a source as reference/ author’s contribution.  

·       English language needs a minor improvement.

(6)   Selection of logistics service mode under direct selling channel+reselling channel mode

·       Before subsection 4.1, a small paragraph can make it better.

·       Section heading should be checked for modification.

·       All paragraphs have a good explanation.

·       It is a well explained with sub sections displaying much clarity of result.

·       Sub sections are too well organized.

·       There is a need of citations.

·       The study has enough tests making the study more robust.

·       Theorems are very well structured.

·       Proof of theorems are a good learning for readers.

·       English language needs a minor improvement.

(7)   Selection of logistics service mode under the mode of direct selling channel and agency channel

·       Before subsection 5.1, a small paragraph can make it better.

·       Section heading should be checked for modification.

·       All paragraphs have a good explanation.

·       It is a well explained with sub sections displaying much clarity of result.

·       Sub sections are too well organized.

·       There is a need of citations.

·       The study has enough tests making the study more robust.

·       Theorems are very well structured.

·       Proof of theorems are a good learning for readers.

·       English language needs a minor improvement.

(8)   Numerical Analysis

·       This section is very informative.

·       Figure 2. needs a source as reference/ author’s contribution.

·       Figure 3. needs a source as reference/ author’s contribution.

·       All paragraphs have a good explanation.

·       This section needs elaboration.

·       This section can be the part of earlier section.

·       There is a need of citations.

·       English language needs a minor improvement.

(9)   Conclusion

·       It is enough as conclusion having a very good elaboration.

·       The presentation with bullet headings can make the conclusion more clear and attractive. 

·       There can be some citations.

·       Line no. 536-559 is a very long paragraph.

·       It is better to present as several paragraphs to increase the credentials.

·       English language needs a minor improvement.

(10) References

·       There must be the increased references adding citations throughout the research.

·       Referencing style must be checked and kept as per the journal guidelines.

Author Response

(The authors gave the same response as above.)

Round 2

Reviewer 1 Report

I appreciate the authors' efforts to consider my comments.

Reviewer 3 Report

Dear editor, 

The authors have addressed my all concern.

Thanks